# Lysosome-Disrupting Agents in Combination with Venetoclax Increase Apoptotic Response in Primary Chronic Lymphocytic Leukemia (CLL) Cells Mediated by Lysosomal Cathepsin D Release and Inhibition of Autophagy

**DOI:** 10.3390/cells13121041

**Published:** 2024-06-15

**Authors:** Madhumita S. Manivannan, Xiaoyan Yang, Nirav Patel, Anthea Peters, James B. Johnston, Spencer B. Gibson

**Affiliations:** 1Department of Oncology, Faculty of Medicine and Dentistry, University of Alberta, Edmonton, AB T6G 2R3, Canada; msubram1@ualberta.ca (M.S.M.); xiaoyan3@ualberta.ca (X.Y.); nkpatel1@ualberta.ca (N.P.); anthea.peters@albertahealthservices.ca (A.P.); 2Cross Cancer Institute, Alberta Health Services, Edmonton, AB T5J 3E4, Canada; 3CancerCare Manitoba Research Institute, Hematologist/Oncologist, CancerCare Manitoba, Winnipeg, MB R3E 0V9, Canada; jjohnsto@cancercare.mb.ca; 4Department of Internal Medicine, University of Manitoba, Winnipeg, MB R3T 2N2, Canada; 5Department of Biochemistry and Medical Genetics, Faculty of Medicine, University of Manitoba, Winnipeg, MB R3T 2N2, USA

**Keywords:** CLL, lysosomes, cell death, cathepsins, autophagy

## Abstract

Venetoclax and obinutuzumab are becoming frontline therapies for chronic lymphocytic leukemia (CLL) patients. Unfortunately, drug resistance still occurs, and the combination could be immunosuppressive. Lysosomes have previously been identified as a target for obinutuzumab cytotoxicity in CLL cells, but the mechanism remains unclear. In addition, studies have shown that lysosomotropic agents can cause synergistic cell death in vitro when combined with the BTK inhibitor, ibrutinib, in primary CLL cells. This indicates that targeting lysosomes could be a treatment strategy for CLL. In this study, we have shown that obinutuzumab induces lysosome membrane permeabilization (LMP) and cathepsin D release in CLL cells. Inhibition of cathepsins reduced obinutuzumab-induced cell death in CLL cells. We further determined that the lysosomotropic agent siramesine in combination with venetoclax increased cell death in primary CLL cells through an increase in reactive oxygen species (ROS) and cathepsin release. Siramesine treatment also induced synergistic cytotoxicity when combined with venetoclax. Microenvironmental factors IL4 and CD40L or incubation with HS-5 stromal cells failed to significantly protect CLL cells from siramesine- and venetoclax-induced apoptosis. We also found that siramesine treatment inhibited autophagy through reduced autolysosomes. Finally, the autophagy inhibitor chloroquine failed to further increase siramesine-induced cell death. Taken together, lysosome-targeting drugs could be an effective strategy in combination with venetoclax to overcome drug resistance in CLL.

## 1. Introduction

Chronic lymphocytic leukemia (CLL) is the most common leukemia in developed nations [1]. The combination of the B-cell lymphoma 2 (BCL2) inhibitor venetoclax with antibodies against CD20, Obinutuzumab or Rituximab, has been shown to be an effective therapy for patients with CLL both in frontline therapy and after relapse [2]. The combination, however, is bone marrow suppressive and reduces lymphocytes, leading to infections [3]. Furthermore, most patients become drug resistant to this therapy [4,5]. Thus, there is a need to develop new therapeutic strategies to overcome drug resistance in CLL.

Lysosomes are organelles in the cell that degrade cellular compounds such as proteins and lipids. It has been shown that lysosome functions are elevated in many cancers, including CLL [6]. The number of lysosomes in CLL cells is higher than in normal B cells, and changes in sphingolipid metabolism alter the membrane structure of lysosomes in CLL cells [7]. In addition, the lysosome plays an important role in autophagy. Autophagy is a process of cellular degradation that recycles amino acids and lipids and eliminates damaged organelles, such as mitochondria, under stress conditions, which depend upon lysosomes fusing to autophagosomes, forming autolysosomes [8,9,10]. In CLL cells, autophagy has been associated with cell survival and drug resistance [11,12]. This suggests that lysosomes could be an attractive target for inducing cell death in CLL cells.

We have previously demonstrated that the lysosomes are a therapeutic target in CLL [13,14]. CLL cells with altered sphingosine metabolism were found to be selectively sensitive to lysosome membrane permeabilization (LMP) following siramesine treatment [7]. Siramesine is a lysosome-disrupting agent that induces apoptosis in many different cancer cells [15,16,17,18]. In combination with the BTK inhibitor ibrutinib (used in the treatment of CLL), it was shown to have synergistic apoptotic responses in CLL cells [7]. Mechanisms of LMP-mediated cell death include activation of cathepsin proteases and mitochondrial dysfunction, leading to increased reactive oxygen species (ROS) [14,19,20,21,22]. Furthermore, obinutuzumab, often used in combination with venetoclax in CLL treatment, has been demonstrated to induce LMP in CLL cells, contributing to its cell death [23,24], but the mechanism of this cell death remains unclear.

Besides obinutuzumab, there are several lysosome-disrupting agents identified. One of the most potent is siramesine. Siramesine was originally developed as an anti-depression therapy by targeting the sigma-1 receptor [25,26]. It was later discovered to inhibit acidic sphingomyelinase (aSMAse), leading to LMP in cancer cells [27]. We and others have shown that siramasine in combination with kinase inhibitors such as ibrutinib gives synergistic apoptotic responses in cancer cells, including CLL cells [7,27,28,29].

In this study, we will investigate whether siramesine and venetoclax give synergistic apoptotic responses in CLL cells similar to the obinutuzumab and venetoclax combination. In addition, we will determine the mechanism of action of siramesine and venetoclax treatment, leading to synergistic apoptosis in CLL cells.

## 2. Materials and Methods

### 2.1. Drugs and Stimuli

Drugs were dissolved in DMSO and stored at −80 °C in accordance with the manufacturer’s recommendations: siramesine and chloroquine (Sigma-Aldrich, St. Louis, MO, USA SML0976), and targeted therapies venetoclax (Selleckchem, Houston, TX, USA S8048) and obinutuzumab (GA101). Drugs were added directly to cell media at a desired concentration for 24 h or 48 h, as specified prior to cell harvest. Cathepsin inhibitor E-64 (Thermofisher, Mississauga, ON, Canada AAJ62933LB0) was dissolved in DMSO, ɑ-tocopherol (Sigma, St. Louis, MO, USA 258024-100G) was dissolved in ethanol, recombinant human IL-4 protein (R&D systems, Minneapolis, MN, USA BT-004), and recombinant human CD40 ligand (R&D systems, Minneapolis, MN, USA 6420-CLB) were dissolved in PBS, as per the manufacturer’s instructions.

### 2.2. Reagents and Antibodies

Stains were purchased from the following suppliers: trypan blue solution (Sigma, St. Louis, MO, USA T8154), LysoTracker Red DND-99 (Thermofisher, Mississauga, ON, Canada L7528), Dihydroethidium (DHE), MitoSox Red (Thermofisher, Mississauga, ON, Canada M36008), DAPI (Sigma, St. Louis, MO, USA D9542), Annexin V FITC (BD Bioscience, Milptitas, CA, USA 556419), and 7-AAD (BD Bioscience, Milpitas, CA, USA559925).

Primary antibodies for LC3B (Cell Signaling, Boston, MA, USA, 2775S), SQSTM1/p62 (Cell Signaling, 5114S), ACTB/actin beta (Sigma, A5441), BCL-2 (Cell Signaling,15071S), PARP (Cell Signaling, 9542), Cathepsin-D (Thermofisher, PA572182), LAMP-1 (Cell Signaling, 15665), TFEB (Cell Signaling, 83010), and BID (Cell Signaling, 2002S) were used for immunostaining and Western blotting according to the manufacturer’s recommendations.

Secondary antibodies for HRP goat anti-mouse IgG (LiCor, Lincoln, NE, USA 926-80010), HRP goat anti-rabbit IgG (LiCor, Lincoln, NE, USA 926-80011), Alexa Fluor 488 goat anti-rabbit (Thermofisher, Mississauga, ON, Canada, A11008), IRDye^®^ 800CW Donkey anti-Mouse (LiCor, Lincoln, NE, USA, 926-32212), and IRDye^®^ 680RD Donkey anti-Rabbit (LiCor, Lincoln, NE, USA 926-68073) were purchased for immunostaining and Western blotting according to the manufacturer’s instructions.

### 2.3. Culturing Primary CLL Cells

Blood samples from patients with CLL were collected from the hematology laboratory at the Cross Cancer Institute with prior informed consent in accordance with policy from the Research Ethics Board, University of Alberta, Canada. Patient characteristics are described in Appendix A. Peripheral Blood Mononuclear Cells (PBMC) were purified from the whole blood using Lymphocyte Separation Media (Wisent Bioproducts, Saint Bruno, QC, Canada, 305-010-CL). CLL cells represented over 95% of the cells in PBMC. Patients were randomly selected based on availability, and samples were excluded from selection based on receiving previous treatment or low baseline cell viability (<30%). Primary CLL cells were isolated from peripheral blood mononuclear cells (PBMCs). Cells were incubated at 37 °C in 5% CO_2_, and treatments were performed in Hybridoma Serum Free Medium (Gibco, Billings, MT, Canada, 12045-076) and seeded at a final concentration of 1 × 10^6^ cells/mL in 35mm suspension cell culture plates.

### 2.4. Co-Culture and Co-Stimulation Experiments

HS-5 Stromal cells were purchased from ATCC (Manassas, VA, USA, CRL-3611) and are cultured in DMEM with high glucose and 10% FBS. At a concentration of 5 × 10^4^ cells/mL, cells were seeded a day before the treatment in a 35 mm dish (Sarstedt, Numbrecht, Germany) and incubated at 37 °C in 5% CO_2_. CLL cells were added on top of the HS-5 cells in Hybridoma Serum Free Medium at a ratio of 50:1 (CLL:HS-5). For assessing the drug resistance, the cells were treated with 1 µM siramesine and 0.05 nM venetoclax. CLL cells were collected after 24 h and 72 h by washing off and assaying for cell viability.

Cells were treated with 1 µg/mL of CD40 and IL4 for one hour before treatment with the drug combinations. The cells were then pelleted, and Annexin V-7AAD assay was performed.

### 2.5. Western Blot Analysis

Cell lysates were collected at the indicated times and lysed using NP-40 lysis buffer using a complete protease inhibitor cocktail (Sigma, St. Louis, MA, USA 539134). Protein levels were quantified using Pierce BCA kits (Thermofisher, Mississauga, ON, USA, PI23225). Samples were run on 12% acrylamide gels and transferred to nitrocellulose membranes (BIO-RAD, Mississauga, ON, Canada) blocked using a 5% milk solution in tris-buffered saline with 0.1% tween-20. Protein was detected using UVP ChemStudio, Analytikjena, or Image studio, Licor, Lincoln, NE, USA.

### 2.6. Flow Cytometry

For lysosome staining, cells were stained with 100 nm of Lysotracker Red DND-99 for 40 min at 37 °C. Intracellular soluble reactive oxygen species were analyzed by staining with 3.2 µM DHE for 30 min after treatment with drugs for 30 min at 37 °C. Mitochondrial membrane permeabilization was analyzed by staining with 100 nM MitoSox Red. Cells were stained with Annexin V-FITC (BD Biosciences, San Jose, CA, USA) and 7AAD (BD Biosciences) for 15 min at room temperature for apoptosis analysis. The stained cells were diluted in 1x Annexin V Binding Buffer. The flow cytometry data were obtained using an Attune NxT (Life technologies, Singapore) flow cytometer. 

### 2.7. Fluorescent Microscopy

Coverslips were placed on a 35 mm dish and coated with poly-L-Lysin 0.01% (Sigma, St. Louis, MO, USA P4707) for 15 min. Primary CLL cells were seeded on top of the coverslip, and the treatment was performed. Cells were fixed using 4% paraformaldehyde and permeabilized using methanol. They were stained for cathepsin D and mounted on the slide using mounting media containing DAPI. Five images were obtained for each treatment performed on three individual patient samples. Numbers of highly punctuated cells were manually counted, and the resulting values were averaged and assessed with the *t* test.

### 2.8. Statistical Analysis and Software

All graphs were made and analyzed using Graphpad Prism 8 software (version 8.0.2). The statistical significance of the patient sample was determined using a one-way ANOVA. Synergy was assessed by calculating various dose responses through a cell viability assay and generating synergy scores through Synergy Finder+ software (version 6.3). Western blot analysis was performed in ImageJ (1.53K) and Visionworks software (version 11.2). Flow cytometry data were analyzed using FlowJo software (version 10).

## 3. Results

### 3.1. Obinutuzumab (GA101) and Venetoclax Induces Lysosome Disruption

Obinutuzumab (GA101) and venetoclax are used in combination to treat CLL [30]. We found that GA101 in combination with Venetoclax significantly increased apoptosis to 55% compared to GA101 at 10% and venetoclax at 40% apoptosis. (Figure 1A and Appendix A). To determine whether the combination of venetoclax and GA101 induces lysosome membrane permeabilization (LMP), the CLL cells were treated with the drugs for 24 h and stained with lysotracker Red DND-99 (Figure 1B). The fluorescence loss indicates LMP as detected by the flow cytometer. We found that GA101 and venetoclax alone were able to increase LMP from 20% to 25%, but the combination significantly increased the amount of LMP to 60%. To assess whether LMP caused by GA101 leads to cathepsin D release, we stained the CLL cells with a cathepsin D antibody after 24 h of the GA101 treatment. There was a decrease in punctate staining with the cells treated with GA101 compared to the control cells, from 80% to 10% punctate staining (Figure 2A,B). To determine whether this cathepsin release contributes to GA101-induced cell death, CLL cells were treated with a pan-cathepsin inhibitor, E-64. When CLL cells were treated with the cathepsin inhibitor, E-64, GA101-induced cell death was reduced from 20% to 10% in CLL cells (Figure 2C). This indicates that GA101 causes leakage of cathepsin D from the lysosomes, contributing to cell death in CLL cells.

### 3.2. Lysosome Disruptor Siramesine and Venetoclax Treatment Induces Lysosome Disruption

In CLL cells, siramesine induces lysosome membrane permeabilization and is very effective compared to the other lysosomotropic detergents [7]. To determine whether siramesine will be effective in combination with venetoclax, we analyzed LMP and cell deaths in CLL cells. CLL cells were treated with 0.05 nM venetoclax and 1 µM siramesine for 24 h and stained with Lysotracker red DND-99. This showed an increase in LMP after siramesine treatment from 10% to 35%, which was significantly increased in combination with venetoclax to 60% compared to the control and individual drugs alone (Figure 3A). It was previously shown that siramesine and ibrutinib increased cell death in CLL cells [7]. We then treated CLL cells with siramesine and venetoclax and determined the amount of total cell death using a trypan blue exclusion assay. We found that siramesine or venetoclax alone induces cell death to around 35%, but when combined, further increased apoptosis to 60%, as determined by a trypan blue exclusion assay (Figure 3B). This increase in cell death was still evident at 48 h (Appendix A). To confirm whether cells were undergoing apoptosis, we used an Annexin V/7AAD apoptotic assay; we found similar results, showing that siramesine and venetoclax increased cell death at 24 h to 40% and 50%, respectively, but when combined, the amount of cell death increased to 80% (Appendix A). Furthermore, the amount of cell death failed to correlate with mutational status, Rai stage, sex, or whether CLL cells were previously treated (Appendix A). We then evaluated the effect of siramesine in combination with venetoclax on causing synergistic cell death. CLL cells were treated with different doses of siramesine and venetoclax. Synergy finder+ was used for the analysis of the synergy between the two drugs (Appendix A). We found that siramesine and venetoclax, in combination, give synergistic cell death responses in CLL cells. Beyond the synergistic apoptotic response to CLL cells, we determined whether siramesine and venetoclax treatment induce cell death in CLL patients’ T cells. We found that siramesine and venetoclax increased apoptosis in CLL cells but failed to induce apoptosis in T cells (Figure 3C). In addition, normal B cells isolated from a healthy donor were treated with siramesine and/or venetoclax, and the amount of cell death and lysosome membrane permeablization was determined. We found that siramesine and/or venetoclax failed to increase cell death or LMP (Appendix A). This suggests that siramesine and venetoclax can selectively induce apoptosis in CLL cells.

### 3.3. Siramesine Increases ROS, but Its Combination with Venetoclax Increases Mitochondrial ROS

Lysosome-disrupting agents induce cell death through an increase in ROS [17,19,29]. To investigate the effect of siramesine and venetoclax on the induction of intracellular ROS, CLL cells were stained with DHE (a probe to detect superoxide) after 24 h of the drug treatment. Treatment with siramesine effectively increased the ROS from 5% to 30%, and venetoclax treatment increased the ROS from 5% to 38%. The combination treatment increased the ROS to 53% (Figure 4A). Since venetoclax inhibits the anti-apoptotic protein Bcl-2, leading to increased mitochondrial ROS [5], we evaluated whether venetoclax alters Bcl-2 protein levels and increases mitochondrial ROS. We found that venetoclax can reduce Bcl-2 expression levels in CLL cells, but siramesine failed to further reduce Bcl-2 expression (Appendix A). To further determine whether mitochondrial ROS is increased following treatment, CLL cells were treated with siramesine and venetoclax alone or in combination for 24 h and stained with MitoSox Red to detect mitochondrial superoxide by flow cytometry. There was an increase in mitochondrial ROS with siramesine and venetoclax alone (30% and 35%, respectively) but was significantly increased with the combination to 53% (Figure 4B), indicating that siramesine and venetoclax increase mitochondrial dysfunction. To assess whether increased ROS causes cell death, we pre-treated the primary CLL cells with antioxidant ɑ-tocopherol and added the drugs after one hour. The antioxidant ɑ-tocopherol significantly reduced the cell death caused by the combination of siramesine and venetoclax from 71% to 18% (Figure 4C). This indicates that ROS contributes to siramesine- and venetoclax-induced cell death.

### 3.4. Siramesine Releases Cathepsin D from Lysosomes, and Cathepsin Inhibitor Blocks Siramesine-Induced Cell Death

It has been previously shown that lysosome-disrupting agents cause the release of cathepsins from lysosomes [6,16,31]. To determine whether siramesine and venetoclax release cathepsins from lysosomes, CLL cells were stained with Cathepsin D staining after treating them with siramesine and venetoclax alone and in combination for 12 h. We found that siramesine treatment reduced cathepsin D punctate staining from 58% to 23%, whereas venetoclax treatment reduced cathepsin D punctate staining from 58% to 42%. The combination of siramesine and venetoclax further reduced punctate staining to 4% (Figure 5A,B). This indicates that siramesine and venetoclax increase LMP in CLL cells. To investigate whether cathepsin release from lysosomes contributes to cell death, we treated the CLL cells with a pan-cathepsin inhibitor, E-64, and subsequently treated them with siramesine and venetoclax alone and in combination. E-64 was able to protect the CLL cells from siramesine and venetoclax-induced cell death, as the combination-induced cell death decreased from 62% to 33% (Figure 5C). We also found that E-64 reduced siramesine-induced cell death from 40% to 15%. In addition, we found that E-64 blocked caspase 3 and PARP cleavage (Appendix A). Since siramesine and venetoclax treatment increased mitochondrial ROS, we determined whether the combination treatment leads to the cleavage of pro-apoptotic Bcl-2 family member BID and whether cathepsin is involved in its cleavage. We found that full-length BID levels decreased after treatment with siramesine alone and in combination with venetoclax, but BID levels failed to decrease in the presence of the cathepsin inhibitor E-64 (Figure 5D). Taken together, this indicates that cathepsins play a role in siramesine- and venetoclax-induced cell death in CLL cells.

### 3.5. Siramesine and Venetoclax Overcome Microenvironmental Protection with IL4 +CD40L and HS-5 Protection against Apoptosis

The microenvironment contributes to drug resistance. IL-4 and CD40 ligation increases the survival of CLL cells and is found in the CLL microenvironment [32,33,34]. CLL cells were stimulated with IL-4/CD40L, followed by treatment with siramesine and venetoclax. This showed cell death following the treatment with siramesine and venetoclax alone or in combination, up to approximately 60% after 24 and 72 h, respectively (Figure 6A,B). In the presence of IL4 and CD40L, the amount of cell death increased to 41% at 24 h and 39% at 72 h (Figure 6A,B). This was not a significant differene from the combination alone, suggesting that IL-4 and CD40L signaling can only partially overcome the increased cell death caused by the combination treatment, depending on the CLL patient. This was further confirmed by the partial cleavage of caspase 3 (Appendix A).

Besides IL-4 and CD40L treatment, CLL cells co-incubated with stromal cells have a survival advantage and resistance to drug treatment [35]. CLL cells were co-incubated with HS-5 stromal cells and treated with siramesine and venetoclax. The apoptotic cell death caused by siramesine and venetoclax increased to 60%, and in the presence of HS5 cells, the combination-increased cell death remained at approximately 60% (Figure 6C). Similar results were found at 72 h (Figure 6D). This indicates that the combination was able to overcome the protection provided by the HS-5 cells.

### 3.6. Siramesine Blocks Autophagy Flux in CLL Cells

It has been suggested that autophagy (self-eating) provides protection against cell death in CLL cells [11,12]. Since lysosomes fuse with autophagosomes to create autolysosomes, leading to the degradation of cellular compounds and further cell survival [9], we investigate whether siramesine inhibits autophagy in CLL cells. Siramesine causes LMP and could prevent the further fusion of lysosomes with the autophagosomes. To determine whether siramesine prevents the formation of autolysosomes, we determine the co-localization of LC3 that is found in autophagosomes with LAMP1 that is found in lysosomes. We found co-localization of LC3 and LAMP1 in control and venetoclax-treated CLL cells but reduced co-localization in the CLL cells treated with siramesine. Co-localization was prevented due to the disruption of lysosomes (Figure 7A). We also found that the autophagy inhibitor chloroquine blocked LAMP1 and LC3 co-localization under all treatments as a control (Appendix A). In addition, we investigated LAMP1 levels following siramesine and venetoclax treatment. We found a decrease in LAMP1 levels in CLL cells treated with siramesine but not cells treated with venetoclax (Figure 7B). The initiation of autophagosomes requires the lipidation of LC3 (LC3-II). When autophagosomes fuse with lysosomes, LC3II is degraded. We found that siramesine treatment of CLL cells increased LC3-II levels similar to the LC3-II levels following chloroquine treatment, which prevents the fusion of lysosomes into autophagosomes (Figure 8A). The combination of siramesine and chloroquine failed to further increase LC3-II levels. Venetoclax reduced LC3-II levels that were not recovered by siramesine treatment (Figure 8A. Appendix A). This corresponds to venetoclax inhibition and autophagosome formation, as we previously published. Finally, the cargo protein p62 transports proteins to the autophagosome for degradation [9]. During autophagy, p62 levels decreased as the protein was also degraded in autolysosomes. We found that treatment with siramesine fails to decrease p62 levels, suggesting that siramesine is inhibiting autolysosome function (Figure 8A and Appendix A). To determine whether siramesine inhibition of autophagy contributes to cell death, we treated CLL cells with chloroquine (an autophagy inhibitor) alone or in combination with siramesine. We found that chloroquine can induce cell death in CLL cells but failed to further increase cell death in combination with siramesine (Figure 8B). Similarly, chloroquine treatment induces PARP cleavage, but this was not further increased in combination with siramesine treatment (Appendix A). Taken together, this indicates that siramesine inhibits autophagy, contributing to cell death.

## 4. Discussion

Our results indicate that CLL cells are susceptible to lysosome-mediated cell death, and a combination of siramesine and venetoclax can potentiate cell death. In addition, we demonstrated that the combination led to mitochondrial ROS production and inhibition of autophagy. Moreover, siramesine had a little effect on the normal T cells in patients and overcame microenvironmental survival signals. Taken together, lysosome-targeting agents in combination with venetoclax could be an effective strategy to treat CLL.

Among the lysosome disruptors investigated, siramesine remains the most potent, as it was effective at the lowest doses. Siramesine was originally developed as a sigma receptor antagonist for the treatment of depression [26]. It was found to be inactive at altering depression in clinical trials but was later discovered to induce LMP, leading to cell death in cancer cells [16]. The other lysosomotropic agents used are in clinical use; clemastine is used as an antihistamine, whereas nortriptyline and desipramine are used as anti-depressants [13,19]. Recent anti-depressants, pimozide and loperamide, were shown to inhibit acidic sphingomyelinase (aSMAse) and induce LMP in glioblastoma cells [36,37]. In AML, the use of a proton pump inhibitor, hexamethylene amiloride, in combination with venetoclax increased cell death mediated by lysosomes [38]. This suggests that clinically relevant lysosomotropic agents exist and may be effective at inducing cell death. This will be the focus of future investigations.

Lysosomotropic agents induce cell death through increased ROS and activation of cathepsins [20,22,28,39]. We found that siramesine caused the release of cathepsin D from lysosomes, and a pan-cathepsin inhibitor reduced cell death. Lysosomotropic agents such as siramesine activate cathepsins, leading to the degradation of anti-apoptotic proteins in a variety of different cancer cells [6,40,41]. In CLL cells, we demonstrated that VPA and fludarabine in combination increase cathepsin B expression and lysosomal-mediated cell death in CLL cells [39]. Despite the role of cathepsins in these cancer cells, lysosome-generated ROS plays an important role in cell death. Using antioxidants, most lysosome-disrupting agent-induced cell death is blocked [19,29,36]. Furthermore, lysosome-induced ROS leads to mitochondrial damage and increased mitochondrial ROS [14,17]. Since multiple cathepsins are released from lysosomes and we showed a pan-cathepsin inhibitor can block lysosome-mediated cell death, future investigation will be focused on whether cathepsin D or others are essential for lysosome-mediated cell death in CLL cells.

Venetoclax in combination with obinutuzumab has become the frontline treatment for CLL [2,5]. It has been previously demonstrated that obinutuzumab induces LMP in CLL cells [23]. The other clinically used anti-CD20 antibody, rituximab, failed to show LMP or cell death in CLL cells [24,42]. This could be due to obinutuzimab being a Type II antibody and its ability to alter the actin cytoskeleton in CLL cells [24]. Nevertheless, we further showed that obinutuzumab releases cathepsin D from lysosomes, and blocking cathepsins reduces its cytotoxicity in CLL cells.

Autophagy has been described as a double-edged sword, contributing to both cell survival and cell death [43]. We previously found that venetoclax inhibits autophagy flux in CLL cells, contributing to its cytotoxicity [44]. Autophagy inhibitors in combination with venetoclax have shown synergistic apoptotic responses, indicating that autophagy provides protection against cell death. In acute myeloid leukemia (AML), increased mitophagy protects cells from venetoclax-induced apoptosis [45]. Furthermore, autophagy protects cells from the tyrosine kinase inhibitor dasatinib [11,46]. Several autophagy genes are also altered in CLL, indicating that alteration in autophagy plays a role in disease progression [47]. Siramesine has been shown in other cancers to induce autophagy, contributing to cell death [15,25,29]. These differences in outcomes could be due to cellular context, such as solid tumors versus leukemia. Using lysosome-disrupting agents that block autophagy will benefit from combining with venetoclax.

Taken together, our findings provide evidence that lysosomotropic agents such as siramesine in combination with venetoclax selectively induce cell death in primary CLL cells. This synergistic apoptotic response is governed by ROS from both the lysosome and mitochondria and could overcome microenvironmental protection. This provides a rationale to develop lysosomotropic agents for the treatment of CLL.

## Figures and Tables

**Figure 1 cells-13-01041-f001:**
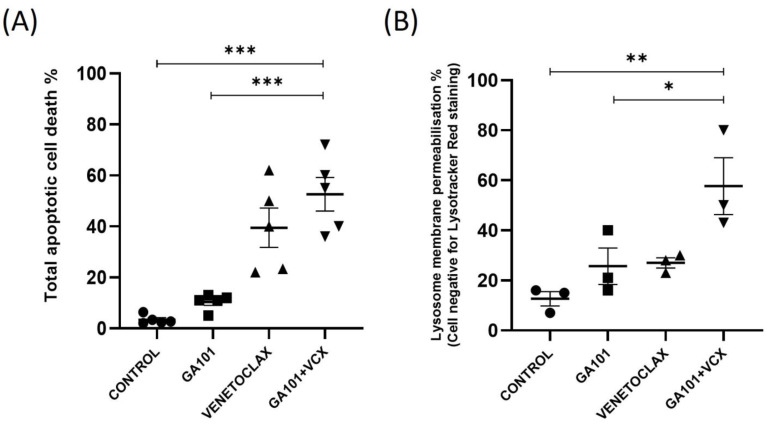
Obinutuzumab synergizes with venetoclax-induced lysosome membrane permeabilization. CLL cells were treated with 50 µg obinutuzumab, 0.05 nM venetoclax alone, and in combination for 24 h. (**A**) The cells were stained for AnnexinV-FITC/7-AAD, and fluorescence was measured using flow cytometry. The graph represents the percentage of total apoptotic cell death (sum of AnnexinV+/7-AAD- and AnnexinV+/7-AAD+) for each treatment condition (N = 5 independent patient samples). (**B**) The cells were stained with Lysotracker red DND-99 for 1 h, and the loss of fluorescence was detected using flow cytometry by gating for the negative population of cells (unstained cells). The graph represents the lysosome membrane permeabilization with each treatment condition (N = 3 independent patient samples). Error bars represent SEM. * represents statistical significance *p* < 0.05, ** represents statistical significance *p* < 0.01, and *** represents statistical significance *p* < 0.001.

**Figure 2 cells-13-01041-f002:**
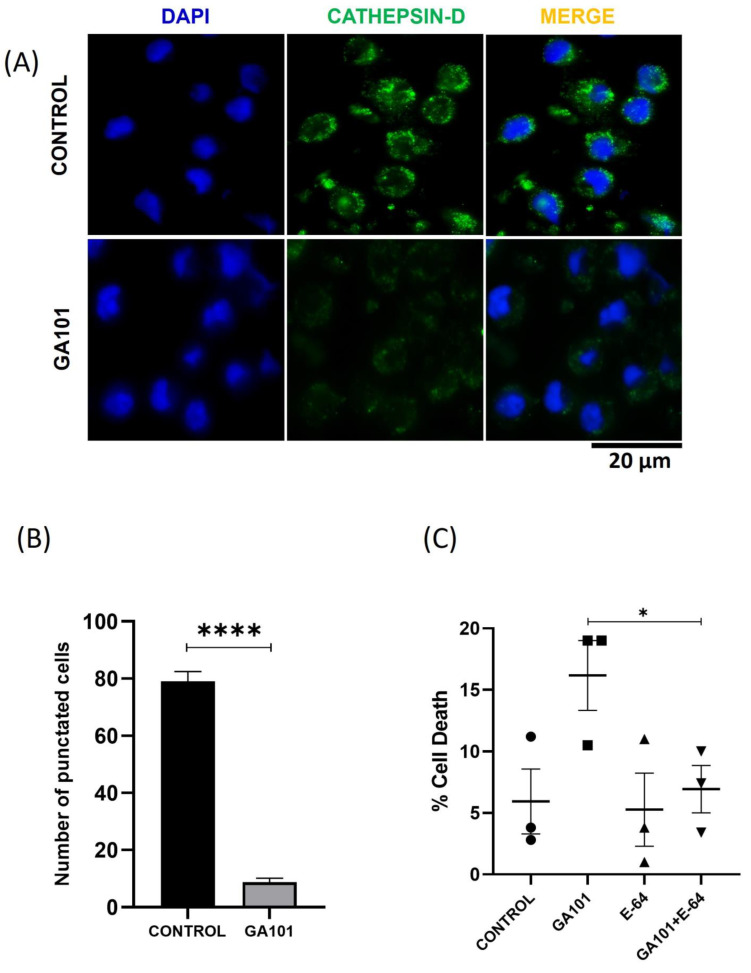
Cell death caused by obinutuzumab is due to cathepsin. (**A**) CLL cells were treated with 50 µg obinutuzumab for 24 h. Representative immunofluorescence staining with an antibody against Cathepsin-D is indicated, and DAPI stains the nucleus. (**B**) The graph represents the number of punctuated cells observed in immunofluorescence images with the control without treatment and obinutuzumab-treated CLL cells. (**C**) The CLL cells were pre-treated with an E-64 pan-cathepsin inhibitor and treated with obinutuzumab for 24 h. A trypan blue exclusion assay was performed after the treatment, and the cell death was quantified. N = 3 independent patient samples. Error bars represent SEM. * represents statistical significance *p* < 0.05, and **** represents statistical significance *p* < 0.0001.

**Figure 3 cells-13-01041-f003:**
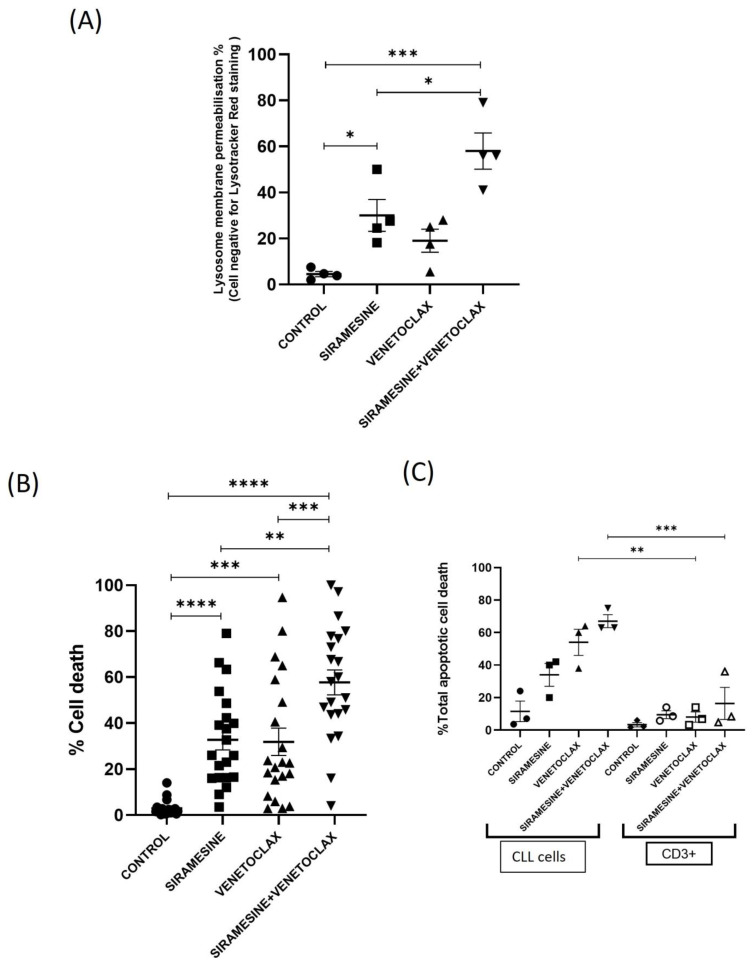
Lysosomal damage-induced cell death is specific to CLL cells. CLL cells were treated with 1 µM siramesine, 0.05 nM venetoclax alone, and in combination for 24 h. The cells were stained, and fluorescence was measured using flow cytometry. (**A**) The graph represents the loss of Lysotracker Red DND-99 with each treatment condition (N = 4 independent patient samples). (**B**) Cell death was determined by trypan blue exclusion assay following treatment. (**C**) AnnexinV-FITC/7-AAD was performed to analyze the total apoptotic cell death of CLL with each treatment condition. The CD3+ was stained prior to AnnexinV/7-AAD treatment and gated for the CD3+ T-cell marker. Error bars represent SEM. * represents statistical significance *p* < 0.05, ** represents statistical significance *p* < 0.01, *** represents statistical significance *p* < 0.001, and **** represents statistical significance *p* < 0.0001.

**Figure 4 cells-13-01041-f004:**
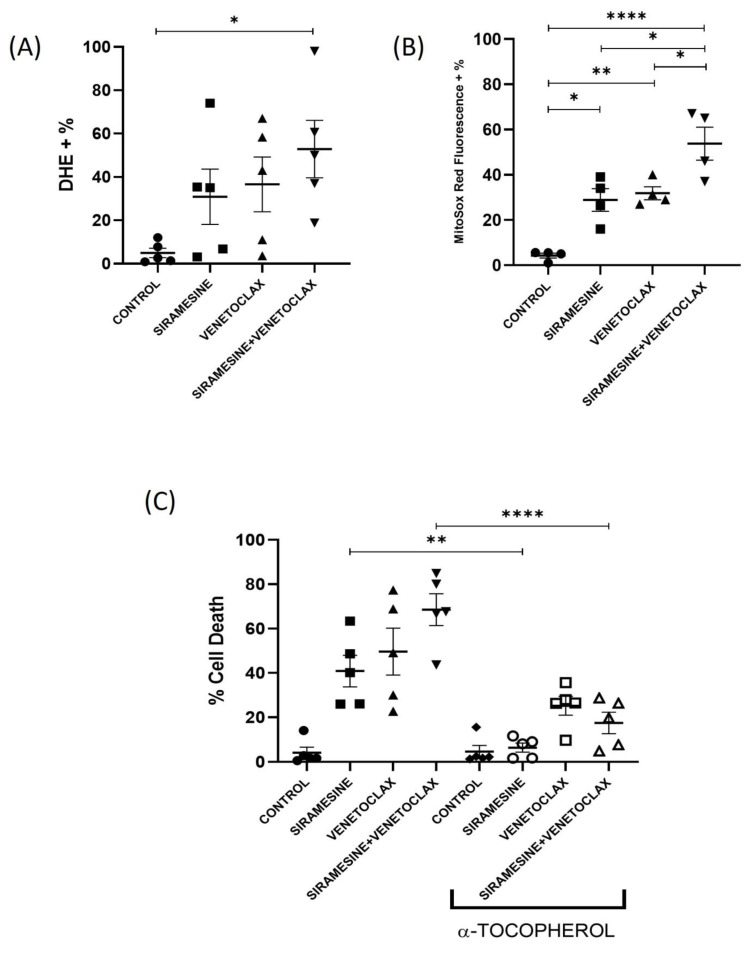
ROS leads to siramesine- and venetoclax-induced cell death. (**A**) CLL cells were treated with 1 µM of siramesine, 0.05 nM of venetoclax alone, and in combination for 24 h. The cells were from five independent patient samples. (**B**) After the drug treatments, the CLL cells were stained with MitoSox Red, and fluorescence was measured using flow cytometry based on control (N = 4 independent patient samples). (**C**) α-tocopherol, an antioxidant, was treated 1 h prior to drug treatments, and a trypan blue exclusion assay was performed to quantify cell death (N = 5 independent patient samples). Error bars represent SEM. * represents statistical significance *p* < 0.05, ** represents statistical significance *p* < 0.01, and **** represents statistical significance *p* < 0.0001.

**Figure 5 cells-13-01041-f005:**
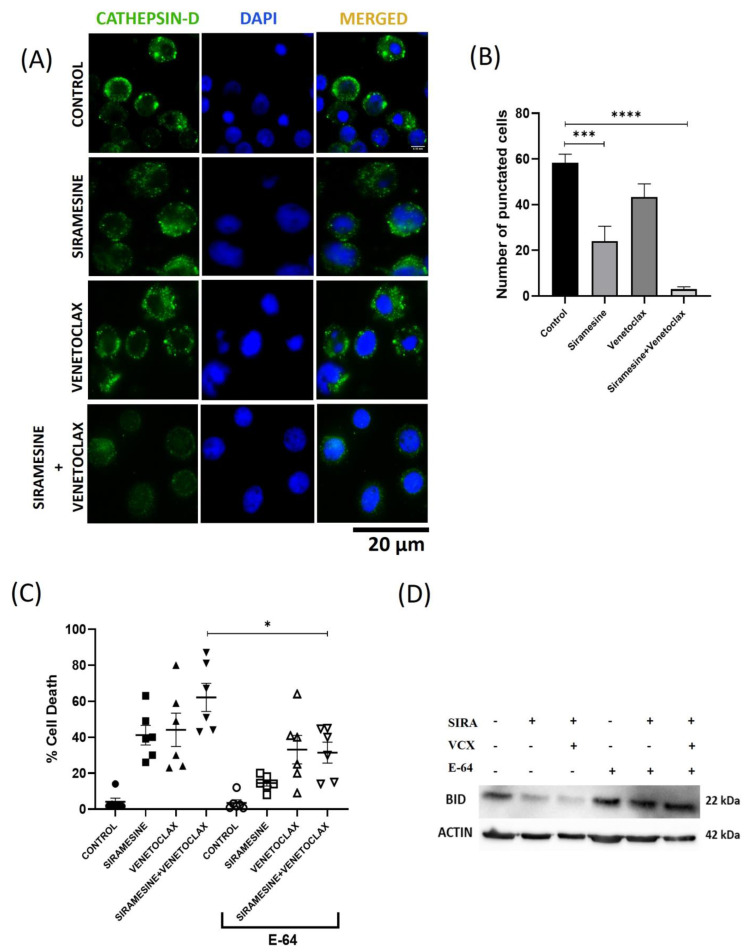
The combination of siramesine and venetoclax leads to cell death through the release of Cathepsin-D. (**A**) CLL cells were treated with 1 µM of siramesine, 0.05 nM of venetoclax alone, and in combination for 12 h. Representative immunofluorescence staining with an antibody against Cathepsin-D is indicated, and DAPI stains the nucleus. (**B**) The graph represents the number of punctuated cells observed in immunofluorescence images with the control without treatment and siramesine and venetoclax-treated CLL cells (N = 3 independent patient samples). (**C**) The CLL cells were pre-treated with an E-64 pan-cathepsin inhibitor and treated with siramesine and venetoclax for 24 h. A trypan blue exclusion assay was performed after the treatment, and cell death was quantified (N = 6 independent patient samples). (**D**) In the cleavage of protein BID, markers for apoptosis were quantified by Western blotting for BID protein. Error bars represent SEM. * represents statistical significance *p* < 0.05, *** represents statistical significance *p* < 0.001, and **** represents statistical significance *p* < 0.0001.

**Figure 6 cells-13-01041-f006:**
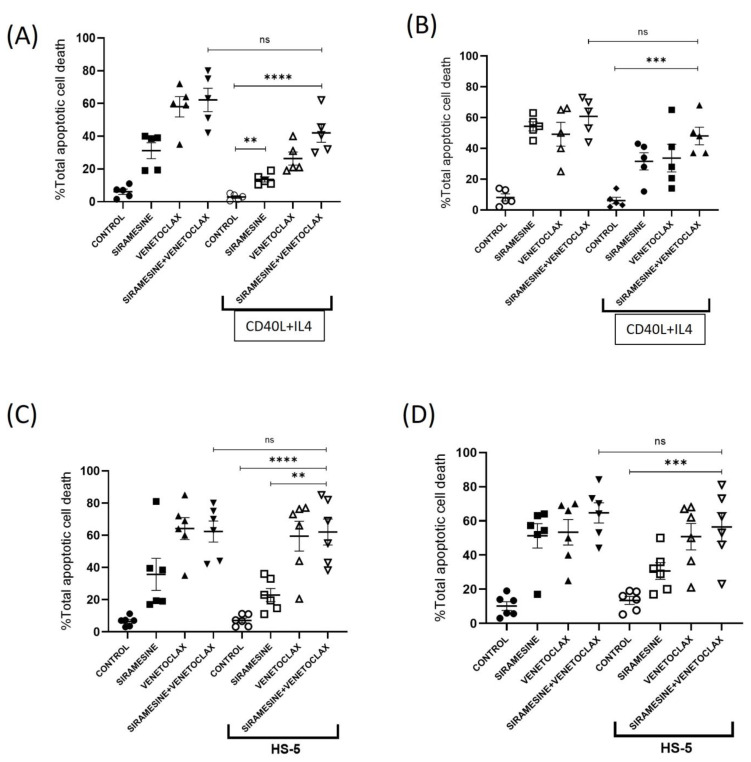
Siramesine and venetoclax overcome microenvironmental protection. CLL cells were pre-treated with CD40 and IL4 recombinant proteins, followed by treatment with 1 µM of siramesine and 0.05 nM of venetoclax alone and in combination for (**A**) 24 h and (**B**) 72 h. The cells were stained for AnnexinV-FITC/7-AAD, and fluorescence was measured using flow cytometry. The graph represents the CLL cells treated with siramesine and venetoclax and the CD40 + IL4 along with drug treatment (N = 5 independent patient samples). CLL cells along with HS-5 cells were treated with 1 µM of siramesine and 0.05 nM of venetoclax for (**C**) 24 h and (**D**) 72 h. AnnexinV-FITC/7-AAD was performed to analyze the total apoptotic cell death of CLL with each treatment condition in the presence and absence of HS-5 cells (N = 6 independent patient samples). Error bars represent SEM. ** represents statistical significance *p* < 0.01, *** represents statistical significance *p* < 0.001, **** represents statistical significance *p* < 0.0001.

**Figure 7 cells-13-01041-f007:**
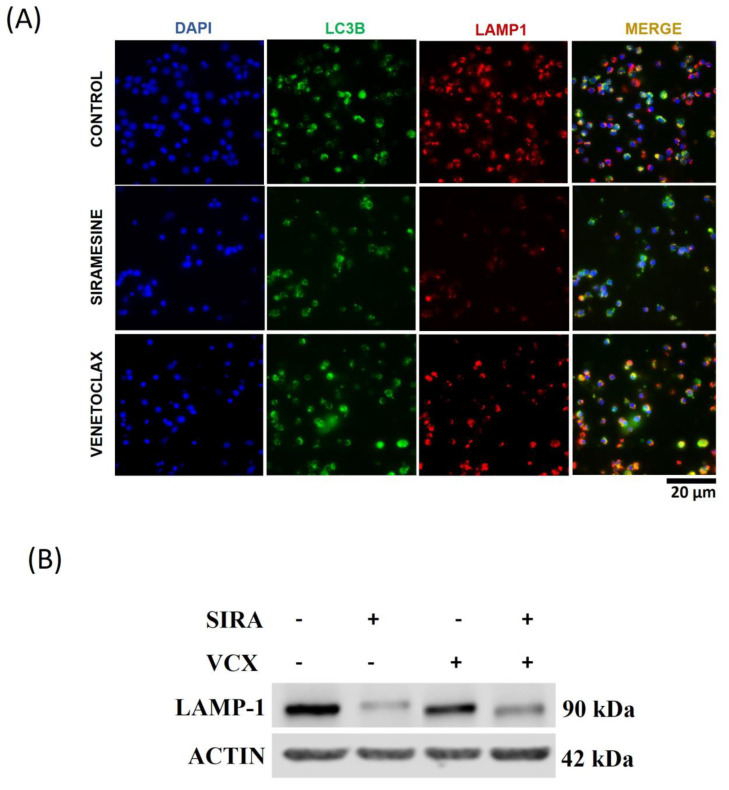
Siramesine inhibits autolysosome formation in CLL cells. (**A**) CLL cells were treated with 1 µM of siramesine and 0.05 nM of venetoclax for 24 h. The representative immunofluorescence of LC3B and LAMP1 is indicated, and DAPI stains the nucleus. (**B**) The marker for the lysosome was quantified by Western blotting for LAMP1 protein. The housekeeping gene actin was used as a control. Error bars represent SEM. This is representative of the three independent patient samples tested.

**Figure 8 cells-13-01041-f008:**
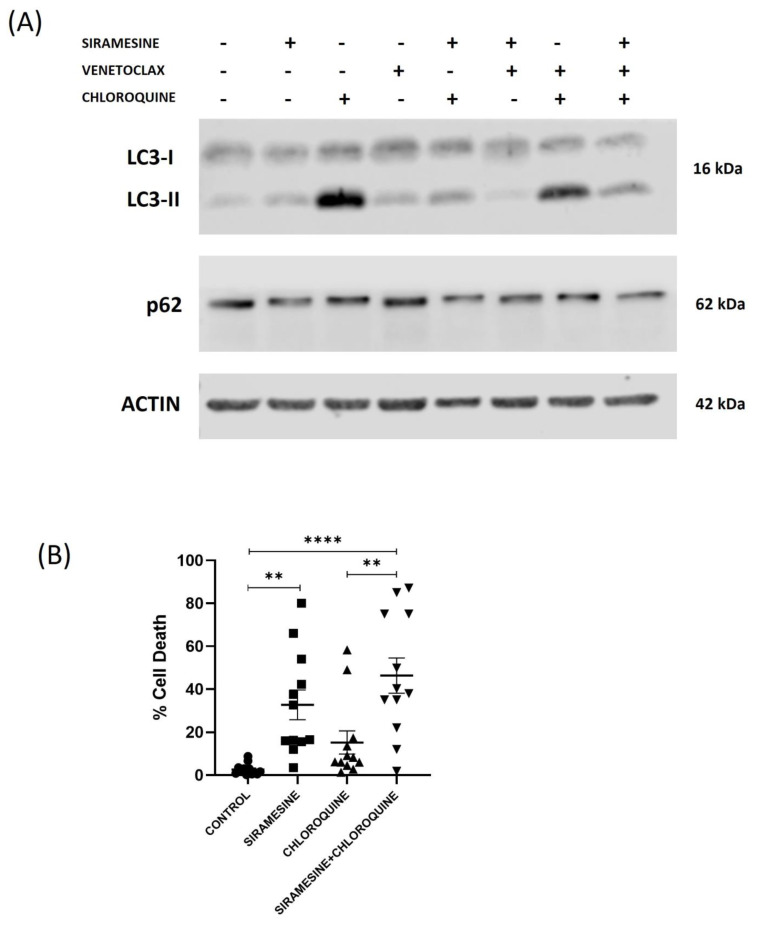
Siramesine inhibits autophagy flux in CLL cells. (**A**) CLL cells were treated with 1 µM of 20 µM of chloroquine and 0.05 nM of venetoclax for 24 h. Autophagy was measured by Western blotting for the autophagy marker protein LC-II in the absence and presence of the lysosomal inhibitor chloroquine. SQSTM1/p62, a protein substrate of autophagy, was also measured by Western blot. This represents three independent patient samples tested. (**B**) A trypan blue exclusion assay was performed after the treatment with siramesine and chloroquine, and cell death was quantified (N = 12 independent patient samples). Error bars represent SEM. ** represents statistical significance *p* < 0.01, **** represents statistical significance *p* < 0.0001.

## Data Availability

No new data were created or analyzed in this study. Data sharing is not applicable to this article.

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
