# Peer review of "Lysosome-Disrupting Agents in Combination with Venetoclax Increase Apoptotic Response in Primary Chronic Lymphocytic Leukemia (CLL) Cells Mediated by Lysosomal Cathepsin D Release and Inhibition of Autophagy"

_cells, 2024, doi:10.3390/cells13121041_

Round 1
Reviewer 1 Report
Comments and Suggestions for Authors
Thank you for sharing the work presented in this manuscript.
Background. Although obinutuzumab is standard of care with venetoclax frontline, it is only being studied in a trial in the second line in relapse. What is more common is the combination of Rituximab in the second line setting based on the Murano study.
What is the impact of rituximab versus obinutuzumab on the findings. This would further help address the choice of obinutuzumab from a clinical standpoint and the head to head data will be of great interest.
The patient characteristics and their responses are not discussed, test there are clear clusters among the samples. Currently the symbols are represented by shapes. What happens if you follow each patient? Any trends related to prognostics markers ? ZAP 70 or mutational status? Any influence of sex ? The N may need to be increased to address this issue.
The western blots are only displayed as a single blot with no graphical analysis of the cohort, please include these in the main figures for all sections so the patient trend across patients is seen.
For the immunoflurescence were any calculations done on the other patients to include with the main figure.
With the various blockers the flow based analysis was reversed what was the impact on the proteins?
The impact of the various combinations on PARP and Caspases by western blot would help tie in the various blockers and the impact of the combinations the pathways being evaluated. This would improve the cohesiveness of the story.
Author Response
Reviewer 1
Background. Although obinutuzumab is standard of care with venetoclax frontline, it is only being studied in a trial in the second line in relapse. What is more common is the combination of Rituximab in the second line setting based on the Murano study.
Response: Thank you for the comment and we have now edited the background to state the combination was used in clinical trial setting and rituximab is more common.
What is the impact of rituximab versus obinutuzumab on the findings. This would further help address the choice of obinutuzumab from a clinical standpoint and the head to head data will be of great interest.
Response: At first glance, rituximab is similar to obinutixumab but when used in vitro, rituximab fails to induce cell death alone or in combination with venetoclax. The antibodies are two different types and have different cellular responses. We added this to the discussion.
The patient characteristics and their responses are not discussed, test there are clear clusters among the samples. Currently the symbols are represented by shapes. What happens if you follow each patient? Any trends related to prognostics markers ? ZAP 70 or mutational status? Any influence of sex ? The N may need to be increased to address this issue.
Response: ZAP70 status is no longer used on a routine clinical basis because it does not direct treatment decisions. Thus, we only 4 patients among the cohort where ZAP70 status is known. Graph based on sex is attached. There were no differences observed between female and male patients (supplementary Figure 2).
The western blots are only displayed as a single blot with no graphical analysis of the cohort, please include these in the main figures for all sections so the patient trend across patients is seen.
For the immunoflurescence were any calculations done on the other patients to include with the main figure.
Response: All the immunofluorescence staining was done on three different patients. The image added to the figures is a representative image. This was added to the figure legend.
The impact of the various combinations on PARP and Caspases by western blot would help tie in the various blockers and the impact of the combinations the pathways being evaluated. This would improve the cohesiveness of the story.
Response: We have now added PARP and caspase 3 cleavage in presence or absence of cathepsin inhibitor (supplementary fig. 6). The inhibitor significantly blocks the cleavage of PARP and caspase 3 in CLL cells.
Reviewer 2 Report
Comments and Suggestions for Authors
Manivannan et al. present an investigation of the effects of obinutuzumab, siramesine and venetoclax in CLL cells. The presented results are mainly either confirmatory or predictable from findings made in previous studies of lysosomotropic agents and venetoclax in CLL and other B-cell malignancies. Although this is not breakthrough research, the study does add some knowledge on the effects of the studied drugs on lysosomal-autophagic processes in CLL cells, and would be worthy of publication after extensive revision.
Important information about the experimental procedures is missing, and there are some errors in descriptions of the results.
The legends to Figs. 2A, 2B, 5A, 5B, 5D, 7A, and 8A and Supplementary Figs. 3, 5 and 6 indicate that experiments were performed on ‘CLL cells’. Were these from a single patient, or from different patients, or was a cell line used?
The Authors need to describe how they evaluated the ‘number of punctated cells’ in the experiments shown in Figs. 2 and 5.
The Results title on lines 173-174 and the title to Figure 1 include the word ‘synergizes’. The experiment shown in Figure 1 did not assess synergy.
The legend to Fig. 3B indicates that 14 patient samples were analyzed. The graph shows 21 data points.
The experiments to assess ROS levels with DHE and MitoSox Red (Figs. 4A and B) would be more informative if the Authors would compare the ratio mean fluoresence intensities/percentages of fluorescent cells. This experiment should have been performed on the same set of patient samples, but the legend indicates 5 for DHE and 4 for MitoSox Red.
On Line 362, the Authors state that the CLL-HS-5 cocultivation experiment was carried out to ‘confirm’ the results of the experiment carried out with IL-4/CD40L. The cocultivation system provides a more complex picture of the contribution of the microenvironment and should be presented in a separate paragraph. It would be appropriate to cite the paper by Seiffert M et al. (2007) that originally described the CLL-HS-5 cocultivation method.
Additional comments/questions
Lines 126-127: how were CLL cells isolated from PBMC?
Line 132: indicate the source of the HS-5 stromal cells (ATCC?).
Line 137 (and several figure legends): The reviewer asks for confirmation that 0.05 nM is the concentration of Venetoclax used in this study- it is considerably lower that the concentration used in several other studies of CLL.
Lines 151-152: the text should specify that DHE is a probe that detects superoxide.
Lines 153-154: MitoSox Red is a probe for mitochondrial superoxide, not for mitochondrial membrane permeabilization.
The description of the comparison of patients’ parameters and in vitro response to the drugs is awkward (lines 237-239). The Authors probably wanted to say that the response to the drugs did not appear to correlate with mutational status, Rai stage or previous treatment. In any case, the large differences in numbers of patients in the different categories make this analysis very weak.
The label for the Y axis in Figs. 1B and 3A reads (Lysosomal Red staining -). It would be more clear to write ‘cells negative for LysosomeTracker staining’.
Lines 397-398: ‘Error bars represent SEM. (N=3 independent patient samples’ This text does not describe Figure 7B (a western blot).
Lines 449-40: ‘N=3 independent patient samples’? The blot shown in the figure appears to show results from one sample.
The legend to Supplementary Fig. 3B indicates N=3 independent patients. Please explain.
On lines 270-288, 316-325, and 357-365, the percentages indicated in the text do not appear to correspond to the positions of the means indicated in the graphs. The text should state exact mean percentages rather than approximations.
The section title on lines 351-352 and the description of the experiments with IL-4/CD40L and HS-5 cells are poorly written.
The Discussion is very weak and would be improved by focusing on how the study’s most relevant observations fit with previous findings.
The phrases on lines 486-487 are confused. The Authors should either clearly describe the relevant findings in the cited publications or delete the text. (For example, in ref. 37, Venetoclax appeared to show LMP activity in an AML cell line.)
Lines 493-494: Venetoclax targets BCL2, not BCL2 family members.
Lines 494-495: ‘The combination of venetoclax and siramesine increasing mitochondrial ROS suggests the importance of the mitochondria in regulating cell death’. This statement is too simplistic and should be deleted.
Comments on the Quality of English LanguageThe manuscript contains many grammatical errors, especially mix-ups between singular and plural and past and present. The Discussion is very poorly written.
Author Response
Reviewer 2
Important information about the experimental procedures is missing, and there are some errors in descriptions of the results.
The legends to Figs. 2A, 2B, 5A, 5B, 5D, 7A, and 8A and Supplementary Figs. 3, 5 and 6 indicate that experiments were performed on ‘CLL cells. Were these from a single patient, or from different patients, or was a cell line used?
Response: CLL cells represent different individual patient sample. The figure legends were revised to reflect this comment.
The Authors need to describe how they evaluated the ‘number of punctated cells’ in the experiments shown in Figs. 2 and 5.
Response: Obtained 5 images from each treatment done on three individual patient sample. Counted the number of highly punctated cells compared to the less punctated ones manually, averaged the values and performed T test. This is now described in the materials and methods section.
The Results title on lines 173-174 and the title to Figure 1 include the word ‘synergizes’. The experiment shown in Figure 1 did not assess synergy.
Response: We are sorry for the wrong choice words and changes synergize to increase in cell death.
The legend to Fig. 3B indicates that 14 patient samples were analyzed. The graph shows 21 data points.
Response: That was a mistake during compiling the data. It is 21 independent patients’ analyzed.
The experiments to assess ROS levels with DHE and MitoSox Red (Figs. 4A and B) would be more informative if the Authors would compare the ratio mean fluoresence intensities/percentages of fluorescent cells. This experiment should have been performed on the same set of patient samples, but the legend indicates 5 for DHE and 4 for MitoSox Red.
Response: Ratio mean fluorescence intensities were calculated by FlowJo software which gave median fluorescence intensity for each individual sample in the experiment performed. To know more on how this was calculated, please find the link attached (https://www.flowjo.com/blog/post/calculating-number-molecules-cells-using-flowjo-v10). Percentages of fluorescent cells are the values obtained after gating the positive population of stained cells by DHE and MitoSox. The graphs represented for DHE and Mitosox are based on the values of the percentage of fluorescent cells obtained after gating for the stained population.
On Line 362, the Authors state that the CLL-HS-5 cocultivation experiment was carried out to ‘confirm’ the results of the experiment carried out with IL-4/CD40L. The cocultivation system provides a more complex picture of the contribution of the microenvironment and should be presented in a separate paragraph. It would be appropriate to cite the paper by Seiffert M et al. (2007) that originally described the CLL-HS-5 cocultivation method.
Response: Thank you for the advice and we have now edited the results based upon your suggestion.
Additional comments/questions
Lines 126-127: how were CLL cells isolated from PBMC?
Response: CLL cells were isolated following ficoll separation. The PBMCs represents over 95% CLL cells in the patients we obtained samples from. We did not further separate the cells.
Line 132: indicate the source of the HS-5 stromal cells (ATCC?).
Response: HS-5 stromal cells were purchased from ATCC Catalog number: CRL-3611
Line 137 (and several figure legends): The reviewer asks for confirmation that 0.05 nM is the concentration of Venetoclax used in this study- it is considerably lower that the concentration used in several other studies of CLL.
Response: We confirm 0.05nM as we need a dose that will allow for increased apoptosis when combined with siramesine.
Lines 151-152: the text should specify that DHE is a probe that detects superoxide.
Response: We have edited the results based upon our comment.
Lines 153-154: MitoSox Red is a probe for mitochondrial superoxide, not for mitochondrial membrane permeabilization.
Response: Thank you and we have edited the results to indicate that MitoSox Red is a probe for mitochondrial superoxide.
The description of the comparison of patients’ parameters and in vitro response to the drugs is awkward (lines 237-239). The Authors probably wanted to say that the response to the drugs did not appear to correlate with mutational status, Rai stage or previous treatment. In any case, the large differences in numbers of patients in the different categories make this analysis very weak.
Response: The reason the number of patients vary in mutational status, Rai stage and previous treatments, is that they are not selected based these characteristics. We will revised the description to state drugs did not appear to be correlated in the results section.
The label for the Y axis in Figs. 1B and 3A reads (Lysosomal Red staining -). It would be more clear to write ‘cells negative for LysosomeTracker staining’.
Response: The Y axis label has been revised.
Lines 397-398: ‘Error bars represent SEM. (N=3 independent patient samples’ This text does not describe Figure 7B (a western blot).
Response: The text has been revised.
Lines 449-40: ‘N=3 independent patient samples’? The blot shown in the figure appears to show results from one sample.
Response: The figure represents the result from three independent experiments. The figure legend has been revised.
The legend to Supplementary Fig. 3B indicates N=3 independent patients. Please explain.
Response: The figure represents the results from three independent experiments. The CLL cells were from three different patients.
On lines 270-288, 316-325, and 357-365, the percentages indicated in the text do not appear to correspond to the positions of the means indicated in the graphs. The text should state exact mean percentages rather than approximations.
Response: The text has been edited to indicate exact percentages.
The section title on lines 351-352 and the description of the experiments with IL-4/CD40L and HS-5 cells are poorly written.
Response: This has been edited to read better.
The Discussion is very weak and would be improved by focusing on how the study’s most relevant observations fit with previous findings.
Response: The discussion has been revised to focus on most relevant observations and how it fits with previous findings.
The phrases on lines 486-487 are confused. The Authors should either clearly describe the relevant findings in the cited publications or delete the text. (For example, in ref. 37, Venetoclax appeared to show LMP activity in an AML cell line.)
Response: We have revised statement to be clearly describe the relevant finding.
Lines 493-494: Venetoclax targets BCL2, not BCL2 family members.
Response: The text has been revised.
Lines 494-495: ‘The combination of venetoclax and siramesine increasing mitochondrial ROS suggests the importance of the mitochondria in regulating cell death’. This statement is too simplistic and should be deleted.
Response: The statement has been deleted.
Reviewer 3 Report
Comments and Suggestions for Authors
This manuscript from Manivannan and co-workers aim to test the response of CLL cells to combination therapies involving lysosomal disrupting agents. This is an interesting concept and can be potentially developed further. Unfortunately, there is no sufficient data as to support the hypothesis and some controls need to be added.
Figure 1. The observed effect could be synergistic or additive, and therefore, a similar analysis such as the one used in Supplementary Figure 3 with SynergyFinder+ should be performed.
Figure 2. E-64 inhibits a range of enzymes, and therefore, to test if the observed effect on apoptosis is due to its inhibitory effect on Cathepsin D an especific depletion of the protein via siRNA transfection or similar must be performed. A similar comment applies to figure 5.
Figure 5. It is recommended to use additional apoptosis markers in western-blot, such as cleaved caspase 3.
Figure 7/8. A more quantitative experimental system should be used to assess the autophagic flux and the autolysosome formation. For instance the use of a GFP-RFP-LC3 reporter and the quantification of the green dots vs red dots and the co-localized dots is a more appropriate system. In addition, controls with BafilomycinA1 (or similar) should be added to all conditions tested. The WB of p62 and LC3B in figure 8A is missing a control adding Siramesine+Venetoclax with CQ.
Comments on the Quality of English LanguageN/A
Author Response
Reviewer 3
Figure 1. The observed effect could be synergistic or additive, and therefore, a similar analysis such as the one used in Supplementary Figure 3 with SynergyFinder+ should be performed.
Response: This is good suggestion, but Obinutuzumab does not increase cell death on a dose depend manner (see below) and thus makes the determination if it acts synergistic and additive not possible.
Figure 2. E-64 inhibits a range of enzymes, and therefore, to test if the observed effect on apoptosis is due to its inhibitory effect on Cathepsin D a specific depletion of the protein via siRNA transfection or similar must be performed. A similar comment applies to figure 5.
Response: We consider knocking down cathepsin D in primary CLL cells but the rate of transfection is very low in these cells and because they primary cells, we cannot select for cathepsin D knockdown cells.
Figure 5. It is recommended to use additional apoptosis markers in western-blot, such as cleaved caspase 3.
Response: We have added cleaved caspase 3 data to supplemental figure 6 and 7.
Figure 7/8. A more quantitative experimental system should be used to assess the autophagic flux and the autolysosome formation. For instance the use of a GFP-RFP-LC3 reporter and the quantification of the green dots vs red dots and the co-localized dots is a more appropriate system. In addition, controls with BafilomycinA1 (or similar) should be added to all conditions tested. The WB of p62 and LC3B in figure 8A is missing a control adding Siramesine+Venetoclax with CQ.
Response: We have now added the additional controls to the experiments in Figure 7 and 8. For the same reason for siRNA transfections, it is not feasible to express GFP-RFP-LC3 in primary CLL cells due to poor transfection efficiency.

Round 2
Reviewer 2 Report
Comments and Suggestions for Authors
The Authors have addressed this reviewer's main comments.
Comments on the Quality of English LanguageThe manuscript will be suitable for publication after editing for English grammar (especially the Discussion).
Below are some suggested corrections.
Line 132: CLL cells represented over 95%...
Line 141: …were purchased from…
Lines 176-178 would read better as follows:
Five images were obtained for each treatment performed on 3 individual patient samples. Numbers of highly punctated cells and less punctated cells were manually counted; resulting values were averaged and assessed with the T test.
Lines 178-179: delete ‘This is now described in the materials and methods section.’
Lines 473-474: Since multiple cathepsins are…
Line 481: The other clinically used…
Line 483: …being a Type II…
Lines 214, 280, 343, 385, 388, 394, 409, 410: delete ‘different’
Author Response
All suggested corrections have been done.
Reviewer 3 Report
Comments and Suggestions for Authors
The authors addressed all the question raised. My only comment is that in figure 8 all blots should be merged into one, since these seem to be the same samples (based on the actin blot) for both p62 and LC3B. It would be easier for the reader to have figure 8A with all blots and figure 8B with the graph. Please, review the figure legend as well since right now is not accurate.
Author Response
Figure 8 has now been revised.